# Phage Endolysins: Advances in the World of Food Safety

**DOI:** 10.3390/cells12172169

**Published:** 2023-08-29

**Authors:** Amina Nazir, Xiaohui Xu, Yuqing Liu, Yibao Chen

**Affiliations:** 1Shandong Key Laboratory of Animal Disease Control and Breeding, Institute of Animal Science and Veterinary Medicine, Shandong Academy of Agricultural Sciences, Jinan 250100, China; aminanazir11@yahoo.com (A.N.); xmsxuxiaohui@163.com (X.X.); liuiuqing@163.com (Y.L.); 2Key Laboratory of Livestock and Poultry Multi-Omics of MARA, Jinan 250100, China

**Keywords:** food preservatives, antibacterial, biocontrol, applications, food pathogens

## Abstract

As antimicrobial resistance continues to escalate, the exploration of alternative approaches to safeguard food safety becomes more crucial than ever. Phage endolysins are enzymes derived from phages that possess the ability to break down bacterial cell walls. They have emerged as promising antibacterial agents suitable for integration into food processing systems. Their application as food preservatives can effectively regulate pathogens, thus contributing to an overall improvement in food safety. This review summarizes the latest techniques considering endolysins’ potential for food safety. These techniques include native and engineered endolysins for controlling bacterial contamination at different points within the food production chain. However, we find that characterizing endolysins through in vitro methods proves to be time consuming and resource intensive. Alternatively, the emergence of advanced high-throughput sequencing technology necessitates the creation of a robust computational framework to efficiently characterize recently identified endolysins, paving the way for future research. Machine learning encompasses potent tools capable of analyzing intricate datasets and pattern recognition. This study briefly reviewed the use of these industry 4.0 technologies for advancing the research in food industry. We aimed to provide current status of endolysins in food industry and new insights by implementing these industry 4.0 strategies revolutionizes endolysin development. It will enhance food safety, customization, efficiency, transparency, and collaboration while reducing regulatory hurdles and ensuring timely product availability.

## 1. Introduction

Bacteriophages (phages) are natural predators of bacteria and act as bacterial parasites that can lyse them and release their progeny [1,2,3,4]. They have two types of life cycle: lysogenic and lytic. In the lysogenic cycle, phage genome is entered into the host genome after attachment and undergoes replication along with host chromosomes. In lytic cycle, they inject their genetic material into the bacterial cell after attachment then undergo replication and assembly of new phage particles, bacterial cell lysis and release of new phages (Figure 1) [5]. That is why, its antimicrobial activities are linked to lytic phage. Due to the effective results, this technique is rapidly used in several fields such as agriculture, veterinary medicine, food safety, and wastewater treatment [6]. Along with the antimicrobial effect against pathogenic bacteria, phages can provide an effective alternative to traditional disinfectants [7]. Due to surface decontamination property, phages can also be applied as food bio preservatives [8]. Thus, phage preparations against *Listeria monocytogenes* (Listshield TM), *Salmonella enterica* (SalmFresh TM), and *Escherichia coli* (Ecoshield TM) have received generally recognized as safe (GRAS) designation by the Food and Drug Administration (FDA) for direct application to food and are commercially available. Phages have several advantages of their use in industry; however, there are some drawbacks including narrow host range, emergence of resistance, lack of broad spectrum activity, limited shelf life and complex interactions [9]. In this context, virus-encoded proteins (endolysins) with the potential to fight against bacteria have attracted attention. Phage proteins can be extracted and also engineered for deployment as an alternative to whole phage therapeutic applications.

In phage research, endolysins were characterized and their roles in the lytic cycle of phages were discovered [9]. Moreover, endolysin applications and their role as antimicrobial agents were also explored. The terms ‘lysins’ and ‘endolysins’ are frequently utilized interchangeably within the field of microbiology; however, it is noteworthy that ‘lysins’ encompass proteins capable of both internal and external cell wall lysis, while ‘endolysins’ specifically denote proteins responsible for lysing the host cell internally during the later stages of viral replication (Figure 1) [10]. After host cell lysis, endolysins are harvested, purified, and characterized. These proteins hold potential as antimicrobial agents and have applications in food safety and medicine.

Endolysins or murein hydrolases are the hydrolytic enzymes produced by the phages (bacterial viruses) for the degradation of the bacterial cell wall and release phage progeny at the end of the lytic cycle [11]. Notably, various bonds in the peptidoglycan structure of the bacterial cell wall are directly targeted by endolysins [12]. During this lysis progression, a few enzymes and holins are involved. Holins are small proteins (>100 nm in diameter) of phages that can make hydrophobic holes in the bacterial cell membrane and send signals for endolysins to degrade the peptidoglycan [13]. However, endolysins exhibit exolysin activity by degrading the outer peptidoglycan layer of Gram-positive bacteria. Nonetheless, they are unable to degrade the outer membrane in case of Gram-negative bacterial cells [14]. The outer layer of Gram-negative bacterial cells prevents the entry of endolysins. Therefore, researchers needed help with novel approaches using the endolysins against Gram-negative bacterial cells.

Phage endolysins infecting Gram positive bacteria are typically categorized in a modular manner having unique enzymatically active domains (EADs) and cell-wall binding domains (CBDs) [15]. EADs provide the simple enzymatic action that degrades the peptidoglycan structure, while CBD can identify and bind to specific cell wall linked ligand molecules with great specificity [16]. This specificity is one of endolysin’s most valuable and unique property because it limits the devastation of only undesirable bacteria leaving the desired flora unaffected (e.g., starter microbe in a fermented product) [17].

The food industry faces a real threat from food-borne microbes that contaminate and spoil food. These microbes can contaminate food at various stages of production, storage, and distribution, leading to foodborne illnesses with health implications ranging from discomfort to severe complications. Beyond public health concerns, such contamination also carries economic consequences due to product recalls, legal issues, and damage to brand reputation [18]. The most serious threat in the form of pathogens to the food industry are *Staphylococcus aureus* (*S. aureus*), *Escherichia coli* (*E. coli*), *Salmonella* spp., and *Clostridium* spp., during food processing can harm human health and leads to a remarkable reduction in economy [18]. It is an endless challenge for the food industry to control these pathogens. Although good hygiene and good manufacturing practices have been practiced, alarmingly, food-borne diseases have elevated and outbreaks are not decreasing [19]. Hence, developing novel strategies to combat severe food-borne threats in the present era is imperative.

Regarding safety, endolysins are not associated with gene transduction issues nor contribute to the increasing possibility of bacterial resistance [20]. However, even though phage applications have some serious concerns such as gene transduction and spreading phage resistant bacteria, amazingly, endolysins do not cause such problems [15]. Therefore, they can be used for food safety purposes as biocontrol agents. Studies have been published on endolysin’s medical applications; however, they have yet to be actively studied in the food industry [11]. However, we have found all in vitro approaches are time/cost consuming and laborious tasks. Alternatively, given the emergence of advanced high-throughput sequencing methods, it is imperative to create robust computational systems to detected phage virion proteins (PVPs), paving the way for upcoming research endeavors. Within this context, artificial intelligence (AL) and machine learning (ML) encompasses potent methodologies that can facilitate the examination of intricate datasets, enabling the extraction of insights and identification of patterns.

We introduce an up-to-date overview of the engineering strategies for phage endolysins and discuss the recent advances in engineering techniques for their application in the food industry. Basic features, advantages, potential of engineered endolysins, limitations of endolysins, and their potential use in the food industry are discussed. We aimed to provide current status of endolysins in food industry and new insights by implementing these industry 4.0 strategies revolutionizes endolysin development. It will enhance food safety, customization, efficiency, transparency, and collaboration while reducing regulatory hurdles and ensuring timely product availability.

## 2. Primary Features of Endolysins

### 2.1. Structure and Enzymatic Activity of Endolysins

Endolysins are the enzymes that express during the later phase of the lytic cycle of the phage virus. After the phage concludes its lytic cycle inside the host, endolysins degrade the host’s cell wall by hydrolyzing the peptidoglycan of the host to release the progeny virions [21].

Gram-positive infecting phage endolysins have a modular structure with EADs at the N-terminal while a linker attaches CBDs at the C-terminal. Endolysins have enzymatic hydrolysis and substrate identification due to custody of EADs and CBDs. Usually, modular endolysins have one or two EADs at N-terminal and CBD at the C-terminal linked by a flexible region called the linker (Figure 2) [22]. The murein layer of host bacteria is cleaved by EAD of modular endolysins while CBD binds to various epitopes in the cell wall to ensure the catalytic effect of EAD. Structurally, phage endolysins are similar to fungal cellulases with flexible linkers connecting EADs and CBDs [23].

Phages infecting Gram-negative bacteria have various structures in their endolysins. However, mostly Gram-negative infecting phage endolysins are 15–20 kD proteins of a simple globular system containing a single enzymatic domain EADs [24]. Gram-negative infecting phage endolysins that display the globular structure have reversed molecular arrangement, i.e., EADs at C-terminal and CBDs at N-terminal, e.g., *Pseudomonas* endolysin KZ144. *Pseudomonas putida* phage endolysin, called OBPgp279 has two CBDs. CBDs in Gram-positive infecting phage endolysins enhance the enzyme’s substrate affinity while CBDs from Gram-negative infecting phage endolysins demonstrate a broad binding spectrum. In addition to holin-mediated endolysins, signal-arrest-release (SAR) Gram-negative infecting phage endolysins are also explored [25]. Initial periplasmic localization of SAR endolysins occurs before releasing into the cytoplasm for host lysis, e.g., ERA103, Lyz103, and phage P1 [26].

### 2.2. Endolysins: Mode of Action

Bacterial host lysis follows a three-step model. Three proteins are involved in this model: endolysin, holin, and spanin. These proteins work together and cause the breakdown of peptidoglycan and inner- and outer membranes (Figure 3). Gram-negative hosts degrade peptidoglycan in two distinct ways: holin–endolysin and pinholin SAR endolysin. The co-existence of holin protein with endolysins improves their efficacy. First, holin regulated the movement of endolysin toward its substrate. Endolysins gather in the cytoplasm during the growth of phage in bacteria. Eventually, holin proteins diffuse into the cytoplasmic membrane, creating holes that enable the endolysins to pass through and degrade the peptidoglycan. This leads to the lysis process and release of the progeny virions [11]. In the final step of lysis, SAR endolysin can be activated due to membrane depolarization by the pinholin. Previously, it was an assumption that the holin–endolysin system of phages allows them to damage the bacterial cell wall [27]. However, new studies have found another protein named spanin also required for the breakdown of outer membrane [28]. Some phages have signal peptides rather than the holin protein, which directs them towards secretary pathways. Endolysins can act as antibacterial agents when applied from outside to the host as they directly approach the peptidoglycan membrane and carbohydrates of cell wall [29]. Purified endolysin is very efficient and successful against Gram-positive bacteria. Due to this effectiveness, endolysins are biocontrol agents against different bacteria including *S. aureus*, *Listeria monocytogenes* (*L. monocytogenes*), and *Clostridium perfringens* (*C. perfringens*) [30,31,32].

## 3. Impact of Endolysins as Bio-Control Agents in the Food Industry

Bacteria are the common cause of food-borne diseases and the ability of bacteria to adopt resistance against antibiotics is a severe threat to food and pharmaceutical industries field [33]. We need to find alternatives to these drug-resistant antibiotics.

Despite their effectiveness as biocontrol agents, phages pose some challenges in their application as antimicrobial agents in the food industry, such as the need to choose a virulent phage to prevent transduction and the possibility of developing phage resistant bacteria [10]. Therefore, endolysins are being appraised as vital enzymes that raise food safety and lower the risk of food-borne illness [34,35,36]. Studies have also explored the potential of endolysins in food applications. Foods can be directly supplemented with purified endolysins. Moreover, the fermentation bacteria *Lactococcus lactis* and *Lactobacillus* spp. are known to produce and secrete endolysins [37].

Endolysin treatment has also been recommended as a strategy to control pathogenic bacteria in vegetables, dairy and meat, avoiding the antibiotic resistance problem. For antimicrobial agents of vegetables, several endolysins have been suggested as potential antimicrobial agents against *L. monocytogenes* in iceberg lettuce (Table 1). Together with the direct treatment of foods with antimicrobials, the sanitation of biofilms on food-contacting materials in food processing chains is also vital to block cross-contamination of bacteria and guarantee food safety. Studies on endolysin treatment have been performed recently to combat biofilms formed by common biofilm-forming pathogens, including *Staphylococcus*, *Streptococcus*, and *Listeria* which are important pathogens of vegetables, meat and dairy products as described below.

### 3.1. Dairy Products

Numerous research describes the characterization of new endolysins against food-borne pathogens when employed to food and utensils (Table 1). According to studies, food-borne pathogens such as Gram-positive *Streptococcus pneumonia*, *S. aureus*, *L. monocytogenes*, *E. faecalis*, and *C. perfringens* can be controlled by using endolysins. An endolysin PlyV12 has demonstrated high lytic activity against antibiotic-resistant *Enteroccocus faecalis* and *E. faecium* [38]. Another endolysin, *Streptococcus uberis* endolysin Ply700, efficiently destroyed bacteria in milk (up to 81% in 15 min) (Table 1) [39].

Endolysin can be used directly on food products as a biopreservative or in conjunction with current preservation techniques as a food biocontrol option. For instance, phage H5 endolysin specific for *Staphylococcus* species reduced the amount of *S. aureus* in pasteurized milk by 1-log in 60 min by 100% in 4 h (Table 1) [40]. Furthermore, a few studies have shown these endolysins’ safety parameters in the human health [41,42,43]. Endolysins can be used in a variety of raw foods as well as in food-producing accommodations as a promising biocontrol agent.

A combination of endolysins and other antimicrobials has been studied in the past. In food systems, there are only a few reports of synergistic applications. For instance, endolysin LysH5 was used to control the level of *S. aureus* in milk because of its strong bactericidal capability (8-log CFU/mL reduction) [40]. *Staphylococcal* endolysin LysH5 also demonstrated a cooperative bactericidal mechanism with nisin [44]. The milk’s endolysin LysH5 and nisin mixture showed an excellent inhibitory effect for *S. aureus* Sa9 with minimum concentration. In contrast when both LysH5 and nisin were utilized separately, they showed minimum lytic results at the same concentration previously used for the mixture of LysH5 and nisin. Nisin combined with low LysH5 concentrations synergistically achieved complete pathogen inhibition at sub-inhibitory concentrations. Lytic activity of HydH5 was reported against *S. aureus* in milk. The LysH5 endolysin with phage vB_SauS-phiIPLA88, HydH5 virion-associated peptidoglycan hydrolase showed a synergistic bactericidal [45]. However, this endolysin–hydrolase combination has yet to be applied to food. These synergistic systems can enhance food safety with low costs and endolysin concentrations. During a recent study, endolysin LysSA97 that targets *S. aureus* was applied with essential oil in foods, e.g., whole milk, skimmed milk, and lean beef [46]. The improved result was observed in skimmed milk due to the lower fat than other whole milk. Consequently, it proved that the lipid content of the food is directly linked with the anti-bacterial activity. The activity of LysSA97 endolysin was reported in combination with carvacrol (essential oil) in beef and milk against *S. aureus* and the reduction of 4.5-logCFU/mL^3^ was observed [46].

Heat stable enterotoxins released by *S. aureus* are responsible for contamination in food and dairy products. Endolysin LysSA11, which is famous for its host *S. aureus* (methicillin resistant termed MRSA), was practiced in milk and utensils (cutting boards and stainless steel), with a reduction of about 2-log CFU/mL in milk and more than 3-log CFU/cm^3^ in ham [47]. In comparison, total carnage of MRSA bacteria was observed on the surface of utensils within 30 min by using a minimum dose of endolysin [47]. The *streptococcal* endolysin B30 and phage λSA2 showed a synergistic killing effect in the milk [48]. *L. monocytogenes* endolysin LysZ5 demonstrated a healthy reduction of >4-log CFU/mL when introduced in the soy milk [49]. By the application of endolysin treatment, *Listeria* can be controlled in soya milk. *Listeria* has also been shown to be synergistically inactivated by a co-treatment of endolysin and high hydrostatic pressure, especially in pressure-sensitive foods [36]. Nisin with endolysin PlyP100 has also shown a synergistic anti-*listerial* effect in Queso Fresco cheese [50]. PlyP825 showed synergistic bactericidal effect with high hydrostatic pressure against *L. monocytogenes* [51]. Endolysin LysCs4 derived from *Cronobacter sakazakii* for biocontrol of infant formula pathogen and showed 50 μg/mL reduction in OD_590_ was seen over 30 min, demonstrating the peptidoglycan degrading ability of LysCs4 [52]. *C. perfringens* endolysin LysCPAS15 inhibits host cells by up to a 3-log reduction in 2 h at 37 °C, and enhanced green fluorescent protein (EGFP)-fused CBD protein (EGFP-LysCPAS15_CBD1) detects *C. perfringens* within 5 min [53]. CS74L and Ctp1L both endolysins showed activity against *C. perfringens* up to 57.8 ± 1% drop in optical density over 8 min and about 1-log CFU/mL reduction in 2 h, respectively [54,55]. Endolysin DLn1 showed reduced *B. cereus* counts by 4.7-log10 CFU/mL in 24 h [56]. Moreover, λSA2 lysin, B30 lysin showed stronger activity with λSA2 lysin (3.5-log CFU/mL reduction at 100 µg/mL) and ClyR showed more than 2-log CFU/mL reduction within 1 min against *Streptococcus dysgalactiae* (*S. dysgalactiae*) [48,57].

### 3.2. Fruits and Vegetables

Minimally processed vegetables and fruits, when subjected to minimal processing, may become vulnerable to pathogens without any subsequent treatment to eliminate them. This underscores the significance of implementing intervention techniques to ensure the safety. Researchers have explored the use of phages in food processing to mitigate bacterial contamination. Nonetheless, it is important to note that phages display efficacy against specific bacterial hosts and can swiftly develop resistance. Additionally, incorporating phages into food processing might not align with consumers’ preference for additives-free products. In addressing this challenge, the emergence of endolysin as a novel natural food solution has gained attention. A newly isolated endolysin cpp-lys has shown strong activity against *C. perfringens* on lettuce. The bacterial viability on lettuce was significantly lower in the group treated with endolysin than in the PBS group [58], and >4-log of *C. perfringens* J1 were removed within 15 min. The findings of this study demonstrated that endolysin cpp-lys has potential applications in controlling *C. perfringens* in the food industry [58]. Furthermore, two endolysins LysWL59 and LysWL60 from bacteriophage LPST10 were identified. These endolysins utilized for the reduction in *S. Typhimurium* in on lettuce. LysWL59 combined with 0.5 mmol/L EDTA decreased *S. Typhimurium* by 93.03% within 1 h on lettuce. Results showed these endolysins represents a promising agent against *Salmonella* contamination [59]. Moreover, Ply511 and Ply118 when combined with artificially spiked iceberg lettuce and whole cows’ milk have been shown to lessen the quantity of viable *L. monocytogenes* cells (Table 1) [60]. Endolysin Ply500 showed 4-log CFU reduction at 25 °C in 24 h (free or immobilized endolysins) against viable *L. monocytogenes* cells in a spiked iceberg [27].

### 3.3. Meat and Poultry Products

Meat and poultry products are another important contaminants of food-borne pathogens because animal-derived food has a high nutritional content, high water activity, and neutral pH. To combat pathogen in meat or livestock, few studies have determined the bactericidal activity of endolysins in MRSA-contaminated ham. An endolysin LysSA11 showed 3-log CFU/mL bacterial reduction in ham artificially contaminated with MRSA within 15 min at refrigerator temperature (4 °C) [47]. Another endolysin trx-SA1 showed reduction in somatic cells and *S. aureus* numbers after infusion of 20 mg of trx-SA1 in udder quarters [61]. LysRODI showed protective efficacy against mammary infections in mice against *S. aureus* and *S. epidermidis* [62]. LysZC1 demonstrated significant activity against Gram-negative pathogens, it demonstrated the highest activity against *Bacillus cereus*. Moreover, LysZC1 was able to reduce the numbers of logarithmic-phase *B. cereus* by more than 2-log CFU/mL in 1 h and also acted on the stationary-phase culture [63]. Moreover, LysCP28 (10 μg/mL) showed high antimicrobial activity and was able to lyse 2 × 10^7^ CFU/mL *C. perfringens* ATCC 13124 and *C. perfringens* J21 (animal origin) within 2 h. It showed the efficacy of in a food matrix as duck meat was contaminated with *C. perfringens* and treated with endolysin (100 µg/mL and 50 µg/mL), which reduced viable bacteria by 3.2-log and 3.08-log, respectively, in 48 h at 4 °C [51]. Endolysin CP25L can stop bacterial growth up to 4-log CFU compared to a control even after 43 h of incubation [64].
cells-12-02169-t001_Table 1Table 1A list of endolysins shows activity against different bacteria in the food industry.Host BacteriaEndolysinFoodResultsReferences*S. uberis*Ply700milk81% killing (*p* < 0.01) was observed when the inoculum was reduced to approximately 600 CFU/mL.[39]*B. cereus*DLn1LysZC1milkfoodViable bacterial counts could be reduced by 4.7-log10 CFU/mL in 24 h.LysZC1 was able to reduce the numbers of logarithmic-phase *B. cereus* by more than 2-log CFU/mL in 1 h[56,63]*S. typhimurium*LysWL59vegetableReduction of 93% of *S. typhimurium* cells on lettucein 1 h when treated with 2.5 μM of lyswl59 and 0.5 mM edta.[59]LysWL60vegetable*S. aureus*LysH5milkAbout 8-log CFU/mL reduction at 37 °C in 6 h.[40,44]Ply187AN-KSH3bmilkAbout 3-log CFU/mL reduction at 37 °C immediately.[65]λSA2-E-LysO-SH3b, λSA2-E-LysK-SH3bmilkAbout 3-log CFU/mL reduction at 37 °C in 3 h.[35]HydH5Lyso, HydH5SH3b, CHAPSH3bmilkAbout 4-log CFU/mL reduction after CHAPSH3b treatment at 37 °C in 15 min.[66]LysSA97milk, beefBeef Synergistic bactericidal effect with carvacrol.[46]LysSA11milk, hamAbout 4-log CFU/cm^3^ reduction at 25 °C in 15 min.[47]Phi11-481milkShowed strong activity at 2–3 mM CaCl_2_.[34]trx-sa1meatReduction in somatic cells and *S. aureus* numbers after infusion of 20 mg of trx-sa1 in udder quarters.[61]lysRODImeatProtective efficacy against mammary infections in mice.[62]*Clostridial* speciesCtp1LmilkAbout 1-log CFU/mL reduction in 2 h.[54]CS74Lcheese57.8 ± 1% drop in optical density over 8 min.[55]LysCPAS15milk3-log reduction in 2 h at 37 °C.[53]CP25Lmeat4-log CFU compared to a control even after 43 h of incubation.[64]LysCP28duck meat3.4-log CFUs compared with control samples in duck meat after 72 h.[51]Cpp-lyslettuce>4-log of bacteria were removed within 15 min.[58]*C. sakazakii*LysCs4powerded milk50 μg/mL, a reduction in OD_590_ was seen over 30 min, demonstrating the peptidoglycan degrading ability of LysCs4.[52]*S. dysgalactiae*λSA2 lysinB30 lysinmilkStronger activity with λSA2 lysin (3.5-log CFU/mL reduction at 100 µg/mL) than B30 lysin.[48]ClyRmilkMore than 2-log CFU/mL reduction within 1 min.[57]*L. monocytogenes*PlyP825mozzarella milkSynergistic bactericidal effect with high hydrostatic pressure.[51]PlyP100cheese, queso frescoAbout 3.5-log CFU/g reduction at 4 °C in 4 weeks.[50]Ply500iceberg lettuceAbout 4-log CFU reduction at 25 °C in 24 h (free or immobilized endolysins). Reduction in viable *L. monocytogenes* cells in a spiked iceberg.[27]Ply511, Ply118lettuceSignificant reduction in viable Listeria cells in whole cow milk.[60]LysZ5soya milkMore than 4-log CFU/mL reduction in 3 h at 4 °C.[49]

## 4. Development of Hybrid Endolysins by Synthetic Engineering to Combat Food-Borne Pathogens

Phages do not intrinsically “sense” the need of bacterial populations for their reproduction or existence in a conscious manner. Instead, their interactions with bacteria are the outcome of evolutionary processes driven by genetic and biochemical interactions. Bacteria can evolve mechanisms to resist phages, and the coexistence of these entities involves intricate dynamics that continue to be a subject of extensive research. In essence, even though endolysins are potent and efficient, bacteria can evolve countermeasures to resist their action in order to ensure their own survival. Natural endolysins have some limitations, which can be overcome through engineering endolysins including: (a) the presence of food components results in low bactericidal activity and limited host range; (b) ineffective killing of Gram-negative bacteria.

### 4.1. Endolysins Engineering against Gram-Positive Bacteria

The inherent alignment of functional domains within endolysin proteins is often referred to as the natural orientation of linkages. The implications of this natural arrangement of domains within endolysins can be quite substantial. The specific configuration and sequential arrangement of these domains play a pivotal role in determining how effectively the endolysin is able to identify and break down the bacterial cell wall.

One of the most noteworthy characteristics of numerous endolysins is their modular structure, which offers a disruptive quality, opening avenues to tailor endolysins for specific applications. The modular architecture of gram-positive endolysins is composed of multiple EADs and CBDs. This modular nature allows for the swapping of domains from different endolysins, offering a means to adjust specificity, amplify activity, or enhance solubility to align with particular needs or objectives (Figure 4).

#### 4.1.1. Mutagenesis or Truncations

(i) Domain deletion or modification: Endolysins’ antibacterial and CBD activity and host specificity are affected by truncation or deletion of a domain [67,68]. PlyGBS endolysin of *Streptococcus* was mutated for the first time. A truncated endolysin was 20-fold more active than the full-length PlyGBS. Horgan et al. constructed modified version of the LysK (an endolysin from *staphylococcal* phage K), featuring an N-terminal CHAPK domain, a central amidase domain, and a C-terminal SH3b cell wall-binding domain. This design enables CHAPK to break down bacterial peptidoglycan between the tetra-peptide stem and the penta-glycine bridge. Notably, CHAPK exhibited notably greater lytic activity against *Staphylococci* in vitro compared to the natural form of LysK [69]. Study found that CBD removal from some endolysins increased their enzymatic activity [70]. This study elucidated that CBD dependence on EAD is associated with its charge. It has been suggested that negatively charged lipoteichoic acids in the cell wall may interact more with positive-charged catalytic domains if the catalytic domain’s charge changes from negative to positive.

(ii) Directed evolution: Direct endolysin mutagenesis was conducted to improve thermo-stability and lytic activity. This study has explained that direct mutagenesis increased the lytic activity of mutated CD27L endolysin against several *L. monocytogenes* serovars by rearranging conserved residues responsible for hydrogen bonding between catalytic sites in the amidase domain [71]. Mart et al. substituted 15 amino acids into CBD to construct a synthetic endolysin called Cpl-7S [72]. As a result, CBD’s lytic activity was enhanced due to its increased net charge. The scientist have improved the thermal stability of LysF1 by 9.4 °C by mutating only one residue in hydrophobic core [73]. Endolysin properties can also be enhanced through protein structure-based site-directed mutagenesis.

#### 4.1.2. Chimeric Endolysins

(i) Domain swapping: The distinct functions of each endolysin domain allow them to be swapped out or recombined with other endolysins to form chimeric enzymes. The domain shuffling or addition strategies have been demonstrated in several studies to increase catalytic efficiency and extend host specificity [57,74,75]. A variety of food matrices including milk and poultry have benefitted from fusion endolysins that are more lytically active and thermo-stable [74]. Roger showed that the fusion protein CHAPSH3b could kill *S. aureus* in pasteurized milk to undetectable levels after 15 min [75]. In addition, Mao et al. generated a fusion protein consisting of LysK’s CBD (KSH3b) and the EAD of Ply187 (Ply187AN) to enhance the lytic activity of Ply187 in milk [65]. In Swift’s study, the amidase domain of phage GVE2 endolysin was fused to the CBD of *Clostridial* phage CP26F endolysin to generate PlyGVE2CpCWB [76].

(ii) Formation of chimeric enzymes: Chimeric endolysins were constructed and screened using high-throughput methods, bringing together random domain swapping and shuffling. An *E. coli* expression system based on two vectors was developed by Yang et al. for in vitro screening purposes [76]. A novel chimeric endolysin, ClyR, was selected during the in vitro screening platform and formed large, distinct lysis zones in *S. dysgalactiae*. The ClyR lytic efficacy and host spectrum in pasteurized milk was much broader than in *Streptococcus* spp.Various research groups used a similar screening method and some modifications when developing recombinant endolysins targeting *S. aureus*. A SPN1S_lysRz protein, derived from *Salmonella* phage SPN1S, was used to establish a quick screening system by Son et al. [77]. Using this system, four *Staphylococcal* endolysins were randomly swapped between *E. coli* cells to lyse them. A chimeric Lys109 endolysin was selected and analyzed for its lytic efficiency. The lytic activity of Lys109 against *S. aureus* on milk and stainless-steel surfaces was highly improved. Further, endolysins of 12 natural *Staphylococci* were rearranged to create Lee’s library of chimeric endolysins [78]. ClyC is a robust antibacterial endolysin; as an in vivo mouse model demonstrating the protective effects of ClyC against MRSA-induced bacteremia, ClyC exhibited high antibacterial activity in Tris buffer (20 mM Tris-HCl, pH 8.0), milk, and animal blood.

### 4.2. Engineering of Endolysins against Gram-Negative Bacteria

The efficacy of endolysins against Gram-negative bacteria is reduced to some extent due to differences in the cell wall structure. Endolysins cannot penetrate and degrade the PG layer below the Gram-negative bacterial cell wall because the outer membrane forms a barrier (Figure 5) [29]. Researchers have stated some techniques to conquer the hurdle of the outer membrane and lyse the Gram-negative bacteria [79,80] (Figure 5).

(1) Endolysins with intrinsic membrane-passing capabilities: The intrinsic lytic activity of some lysins has been reported against Gram-negative bacteria. The C-terminal region is cationic in these lysins, facilitating contact with negatively charged lipopolysaccharides and helping them to pass through the outer membrane. There is an exception to this observation due to the modular structure of the OBPgp279 lysin from the *Pseudomonas fluorescens* phage OBP. *Pseudomonas aeruginosa* (*P. aeroginosa*) was mildly lysed by this enzyme (~1-log killing). Based on Low’s research, a net positive charge at the EAD could cope with electrostatic repulsion caused by negatively charged bacterial surfaces. However, OBPgp279 contains no amphipathic helixes or positively charged amino acids at N- or C-terminal ends. Further, structural analysis is required to understand how this modular lysin is permeabilized.

It was reported that some phage endolysins could outbreak Gram-negative bacteria naturally without any assistance. Some endolysins have been reported to kill Gram-negative bacteria from the outside of bacterial cells but their lytic activity is inefficient. *Salmonella* phage SPN9CC killed about 2-log CFU/mL *E. coli* cells within one hour. *Acinetobacter baumanni* (*A. baumannii*)endolysin LysAB2 also showed antibacterial activities against Gram-negative bacteria (e.g., *A. baumannii* and *E. coli*) without assistance: amphipathic peptides potentially penetrate the negatively charged outer membrane, but the lytic activity is weak because the extent of limited penetration. A novel lysis mechanism revealed that *Salmonella* endolysin M4Lys was not dependent on either holin. It was observed to have a unique mosaic structure, and the C-terminal transmembrane domain.

(2) Endolysins and outer membrane permeabilizing agents: Previously, chelating agents as the membrane-permeabilizing agents, can be used to enhance the efficiency of endolysins as biocontrol agents for Gram-negative bacteria. For example, EDTA and organic acids have been used as chelating agents. Organic acid (Citric, Malic) treatment provides much better outcomes than EDTA treatment. However, both EDTA and organic acids stay challenging because EDTA is notorious for damaging human living cells and organic acids can deactivate the endolysin in acidic pH conditions. Combining endolysins with physical stressors (e.g., high hydrostatic pressure) also produced significant antibacterial results. High hydrostatic pressure empowers the Gram-negative endolysins to permeabilize and reach their substrate. Endolysin LysSs1 revealed significant antibacterial impacts when pre-treated with heat or chloroform against the Gram-negative bacteria. However, both methods have some safety and practical application issues.

(3) Engineering lysins with OM disrupting properties: Researchers have also revolutionized this fusion technique between Gram-negative bacteria endolysins and peptides (membrane-penetrating peptides) to make endolysins genetically modified. There are three proposed techniques. (1) Fusion of endolysin and polycationic peptide, Endolysins at this genetically modified state are termed as Artilysins. Artilysins’ structure comprises domains of endolysins, linker units and some peptides. Advanced molecular biology skills are required to create genetically modified endolysins. It was shown that *Salmonella* endolysin M4Lys has not relied on holin proteins when related to other endolysins. Artilysin Art-175 is a fusion protein of endolysin KZ144 with SMAP-29 at the N-terminus. It was found that Art-175 had bactericidal activity against *A. baumannii*, *E. coli*, and *P. aeruginosa*, as well as eradicating large inoculas of bacteria (≥10^8^ CFU/mL) [24]. (2) Fusion with bacterial membrane-translocating domains. OM-translocating and endolysin-delivery bacteriocins can be engineered to deliver endolysins to peptidoglycan areas in gram-negative bacteria [81]. Lycosins are fusion proteins formed by combining endolysin and bacteriocin, facilitating membrane transport. (3) Peptides with bactericidal activity formed by truncating lysins called innolysins. Ec21 is an innolysin that showed bactericidal activity and *E. coli* cells resistant to third generation Cephalosporins have increased by as much as three logs [82].

(4) Formulation of endolysins in carrier systems: During the recent work of endolysins against Gram-negative pathogens, a liposome-mediated endolysin encapsulation system was developed due to liposomes’ ability to penetrate the host membrane through membrane fusion. For instance, *Salmonella* endolysin BSP16Lys encapsulated into cationic liposome [83]. As liposomes penetrate bacterial cell membranes by fusion, BSPLys-encapsulated liposomes produced approximately 2.2-log CFU/mL reduction in *Salmonella* without a membrane permeabilizer. The application of endolysins to kill Gram-negative pathogens is still in development as researchers increase their interest in finding novel strategies to control Gram-negative bacteria.

## 5. Application of Endolysins against Bio-Films

Bio-films are the mass of bacterial groups embedded in the extracellular matrix (self-produced matrix containing a complex of polysaccharides, proteins, and DNA). Different growth stages of biofilms have been shown in Figure 6. The control of biofilms in the food industry is very significant as their occurrence may cause serious health hazards [84,85,86]. Troublingly, bacteria in biofilm are more disturbing than planktonic cells [87]. According to research on LysSTG2 as a potential biocontrol agent against biofilms, LysSTG2 can regulate *Pseudomonas* cells in their planktonic and biofilm forms, opening the door to its possible usage in a variety of food products and contact surfaces. In the presence of 5 mmol/L EDTA, 100 g/mL LysSTG2 decreased *P. aeruginosa* and *P. putida* viability by 5.5-log and 3.49-log in 1 h, respectively [88]. Additionally, this combination demonstrated potent antibacterial activity in bottled water against both strains, particularly *P. putida*, bringing viable count values down to the lower detection limit [89]. Due to its strong antibacterial and anti-biofilm action, LysSTG2 may be a contender for use as a biocontrol agent against Gram-negative bacteria in various food processing conditions (Table 2) [90].

Endolysins are promising weapon against biofilms in the food industry compared to other antibiotics. *Staphylococcal* endolysins and derivative proteins removed biofilms formed by *S. aureus* and staphylococcus epidermidis. Phi11 and SAP-2 (*Staphylococcal* endolysins) were used against the biofilms on the polystyrene surfaces [91,92]. In a recent study, endolysin LysH5 was used against the biofilms (*S. aureus* and *S. epidermidis*). A reduction of 1–3 logs was noticed with 6 h of treatment viewing better consequences than lysostaphin. In this study, bacteria surviving the primary LysH5 therapy did not become insensitive or resistant to the endolysin [31]. Furthermore, scanning electron microscopy, safranin staining, and cell reduction have revealed that SAL200 endolysin removes bacteria efficiently [93]. PlyGRCS, another staphylococcal endolysin with a single EAD, can kill MRSA and disrupt biofilms [94]. Additionally, CHAPk, a LysK endolysin truncated only to have the N-terminal endopeptidase domain, can remove biofilms from surfaces caused by *S. aureus* [95]. Researchers found an endolysin LysCSA13 from staphylococcus bacteria effective in removing staphylococcal biofilms from polystyrene, glass, and stainless-steel surfaces. Additionally, biofilm mass was reduced by 80–90% [96]. LysSMP, a *Streptococcus suis* endolysin, disrupted >80% of biofilms formed by 32 different biofilm forming strains of bacteria. These results are proportional to reductions from antibiotics or phages [97]. Biofilm matrixes are rapidly degraded by *Streptococcus pyogenes* endolysin PlyC. The PlyC treatment rapidly destroyed the biofilm matrix, even though streptococcal cells within the biofilm became resistant to antibiotics [98]. Bio-film formation on abiotic surfaces was inhibited by the amidase domain of endolysin extracted from *L. monocytogenes* phage vB_LmoS_293 [99]. Lys68 from *Salmonella* reduced biofilms by approximately one log CFU when combined with malic or citric acid [100]. Moreover, the biofilm structure was destroyed by *P. aeruginosa* endolysin LysPA26, and the viable count was reduced by 1- to 2-log CFU (Table 2) [101]. Endolysin PlyLM possessed disrupting ability against *Listeria* biofilms and showed the synergistic effect along with protease [102].
cells-12-02169-t002_Table 2Table 2List of endolysins applied against biofilms.Host BacteriaEndolysinResultsReferences*S. aureus*lysH5Notable staphylococcal biofilm removal activity against persister cells obtained after treatment with rifampicin and ciprofloxacin.[31]lysCsa13Reduction in staphylococcal biofilms mass up to 80–90% on various food utensil surfaces, including polystyrene, stainless steel, and glass.[103]*S. pyogenes*PlyCDestruction of biofilm matrixes of *S. pyogenes* which showed rapid resistance to traditional antibiotics.[98]*Salmonella*lys68Synergistic biofilm-reducing effect in combination with malic or citric acid by 1-log CFU.[100]*P. aeruginosa*, *P. putida*lysPa262–3 log reduction in viable biofilms of *P. aeruginosa* 8327 on a polystyrene plate for 48 h.[101]LysSTG25 mmol/L EDTA, 100 g/mL LysSTG2 decreased viability by 5.5-log and 3.49-log in 1 h.[88]*Listeria*PlyLMDisrupting ability against *Listeria* biofilms, synergistic effect with a protease.[102]

Based on these studies, endolysins are feasible anti-biofilm agents that help reduce biofilm formation in the food industry. However, more realistic settings would be necessary for evaluating endolysin’s anti-biofilm activities, especially in biofilm matrixes and surface coatings in food settings is necessary to investigate [104].

## 6. Impact of Food Matrix on the Lytic Activity of Endolysin

Laboratory research has demonstrated endolysins to be potent antimicrobials. However, their performance in food settings may differ substantially from their in vitro performance since food is a giant biometric complex with its characteristics. Food systems incorporate some unique parameters such as carbohydrate percentage, pH value, temperature stability, ionic composition, and protein percentage during the processing and storage of food commodities. Optimal conditions must be evaluated before using endolysins in food systems. Temperature or thermo-stability is a crucial parameter when considering endolysin’s efficacy. Researchers have examined how endolysins can be genetically engineered to provide high a thermo-stability [10].

Effect of Ionic concentration: *Listeria* phage endolysin Ply500 having one EAD and two CBDs after engineering enhanced the lytic activity of Ply500 endolysin even at the higher ionic concentrations [105]. At normal ionic levels, non-engineered endolysin had six-fold improved lytic performance. At high ion concentrations, generally 1500 mM NaCl or above, engineered endolysin shows superior activity having two-fold enhanced lytic activity than the native endolysin. Single CBD showed peak efficiency at 150 mM, but beyond 150 mM endolysin and phage significantly reduced the lytic activity [105]. Efficacy in the lytic activity of endolysin was shown to be dependent on different ions having different ionic concentrations [44]. Zn^2+^ and Mn^2+^ showed significant reduction in the lytic activity of LysH5 endolysin while calcium, magnesium and NaCl showed improved lytic activity of LysH5 [44].

Further, LysSA11 was characterized by examining its metal ion dependence. EDTA-treated LysSA11 had significantly less activity than the control; however, Ca^2+^ supplemented LysSA11 had approximately 150% more activity than the control. The specific calcium dependence of this endolysin makes it worthwhile to characterize other endolysins having CHAP domains in a similar manner. For example, LysH5 and Sk1 possess CHAP domains [46,50].

Effect of salt, pH, and temperature: *L. monocytogenes* phage P100 endolysin PlyP100 (N-acetylmuramoyl-_L_-alaline amidase) showed satisfactory inhibitory outcomes against various strains of *L. monocytogenes* [106]. Van Tussell studied the effects of pH (3 to 11), temperature (between 4 to 50 °C), and the salt ratio of NaCl (0 to 500 mM) on endolysin activity. Buffer phosphate revealed enhanced lytic activity of PlyP100 (70–80% achievement) at neutral or near-neutral pH, having 150–150 mM concentration of NaCl wand temperature range of about 37 to 50 °C [106]. These studies concluded that pH, salt, temperature, and food matrix directly impact the degree of endolysin efficacy. Therefore, understanding their optimal conditions is very important before applying endolysins to foods.

Another endolysin LysSA11 from *Staphylococcus* phage SA11 was investigated for different ranges of pH (2 to 10), temperature (4 to 65 °C) and salt concentration (0 to 300 mM) [47]. It was found that pH 8, 200 mM NaCL and 37 °C were optimal for this endolysin. The result suggested that endolysins LysSA11 and LysH5 are suitable candidates for foods such as ham and milk, which have moderate to high sodium salts [47,50]. Further, *S. aureus* concentration was significantly reduced by 1.44-log CFU/mL when LysSA11 was added in ham and milk and incubated at 4 °C. Results suggest that endolysin LysSA11 reduces *S. aureus* numbers in food types and at different temperatures [47].

A broad spectrum of lysis activity was observed for Spp64 and Spp62 to be lysing-related proteins of phage Spp001 [107]. We measured the influence of temperature on Spp64 by placing it in a water bath at different temperatures (4–100 °C) before adding it to *S. putrefaction*. The protein perform effectively at various temperatures [107]. Their lytic activity was higher at cold temperatures (approximately >90% at 4, 30, 40, and 50 °C), and the lytic effectiveness decreased by about 10% after 30 min at 70, 90, and 100 °C [107]. *Shewanella* Sp225 cells were suspended in pH ranges between 3.0 and 10.0 to determine the optimal pH. The optimal pH was 7.0 (nevertheless 80% activity was maintained at a pH range of 5–10) [107].

The optimal administration protocol for endolysins within the food production process also depends on knowledge of its characteristics, including pH, temperature, salt, and ion concentrations [10]. LysH5 was ineffective against *S. aureus* at a high temperature of 37 °C but showed good activity at 4 °C. Therefore, it would be better to add it after the milk or other foods have been heated or thermally processed [10,40]. It is therefore important to understand these characteristics before administering an endolysin, as they may help determine the suitability of an endolysin for the food matrix and the best time for its administration.

Effect of preparation methodology of endolysin: Endolysins are globular proteins used as antimicrobials in food. To utilize them efficiently, pH and cofactor properties should be considered because food systems differ significantly regarding pH and molecular composition [108]. The best pH and temperature for Sk1 (N-acetylmuramoyl-L-alanine amidase enzyme) were 6.5 and 30 °C, respectively. Skl became less effective (20 mM sodium phosphate buffer, pH 6.5) by dialysis in the protein purification process, in contrast to other endolysins [108]. This situation shows dependence on a weekly associated cofactor. It was necessary to pre-incubate Skl for five minutes at 0 °C with choline-containing cell walls to achieve the full enzymatic activity. A total of 2% choline chloride inhibited enzyme activity as a choline binding protein [108]. In conclusion, endolysin preparation methodology and food matrix composition can significantly influence the effectiveness of endolysins, and these factors must therefore be considered when using endolysins. Identifying cofactors that can enhance or reduce the efficacy of endolysins and other enzymatic biocontrols can be very valuable.

## 7. Advantages and Limitations of Endolysins in the Food Industry

Numerous independent experiments have demonstrated the efficacy of endolysins as anti-bacterials [11]. It was immediately recognized that they had several advantages and limitations after they were conceptualized as antibacterial agents (Figure 7). The unique properties of endolysins that make them different from other anti-bacterials in the food industry are their specificity, unique mode of action, synergism, and no development of resistance. Most CBDs are located at the C-terminus and recognize cell wall-associated ligand molecules with remarkable specificity when interacting with their substrate. Due to their specificity, endolysins have numerous advantages from an application perspective, because they can be used directly to target undesired pathogens while leaving a potentially beneficial background flora intact.

As well as being safe, endolysins do not evoke resistance as they have specific action sites which are challenging to mutate and a limited number of potential resistance mechanisms against agents attacked on the cell wall. Endolysins or Artilysins were repeatedly introduced into strains to check their ability of resistance development, but did not produce resistant mutants compared to control antibiotic exposure. Therefore, endolysins and Artilysins are low-risk due to low resistance development rates compared to traditional antibiotics. Further, they can engineer tailored antimicrobials due to their modular design.

In addition to traditional antibiotics, endolysins have been demonstrated to synergize with other endolysins. As a result, the enhanced destructive effect within the three-dimensional PG network is explained by the synergistic effect of endolysins with different catalytic specificities. Another possibility is that one endolysin could cleave the first bond and make it easier for the other to reach the second target site after the cleavage of the first bond. Consequently, the substrate may degrade faster.

Endolysins have the potential to perform a remarkable role in the biopreservation of food. Hence, phage-derived enzymes still face concerns despite the considerable food industry demand for continued studies in related challenges. We need to address them before using the endolysins in the food industry as biocontrol agents such as safety, stability, efficiency, sensitivity, production cost, and regulatory issues.

Developing large-scale industrial endolysins production at low cost, fast, and efficient is a serious issue that should be addressed further. There are well-established protocols for enzyme production in laboratories, but laboratory data are only sometimes appropriate for large-scale productions [109]. A key drawback for all protein-based treatments is their cost, which is typically a barrier to their actual industrial adoption because of their comparatively expensive production, purification, storage, and supply chain costs compared to their conventional synthetic-based alternatives. For endolysin purification and expression, a cost-effective and efficient expression system is necessary for the rational recombination engineering [29]. Endolysins are therefore expected to be more commercially appealing as antimicrobial agents if their solubility and expression levels are improved, as well as cost-effective expression methods. From the onset of protein expression to downstream processing, many ways exist to reduce costs and increase safety. After the genetic engineering of endolysins, down streaming processes including cell lysis, harvesting and purification were performed. It is imperative to remove endotoxin from bacteria when expressing proteins. Further, to achieve high expression levels at low costs, optimal hosts for expression, e.g., bacteria or yeast are important. Plants have been suggested to offer a low-cost and safer production vehicle for recombinant proteins. This study has found no disease symptoms in transgenic tomatoes despite bacteria not being completely removed as these tomatoes express an endolysin CMP1 [110]. Plant expression and endolysin production remain challenging due to the safety concerns surrounding transgenic food products. Moreover, endolysins released through fermentation processes may also reduce treatment costs. Bacteria belonging to *Streptococcus*, *Enterococcus*, *Lactococcus,* and *Lactobacillus* are GRAS. Many endolysins from these bacteria are safe and effective in food systems as biopreservatives including Ply511, CP25L, and LysSA11.

Endolysins have proteinaceous nature and can quickly be affected by environmental factors. Their antibacterial efficacy varies according to the food’s proteins, carbohydrates, fat, minerals, vitamins, and biochemical elements such as temperature, pH, and ionic strength. An acidic environment can disrupt the structure of certain endolysins as endolysins have cleavage sites. Additionally, endolysin structures, linker regions, charge distributions, cofactor usage, and affinity are aspects of endolysin structures that can provide insight into understanding how individual endolysins influence foods. So, structure analysis of endolysins with the help of advanced bioinformatics techniques should be performed to increase the antibacterial activity of endolysins in different foods.

Regarding regulatory issues, recombinant DNA technology and heterologous expression are the most common strategies used in the studies of endolysins-based treatment. There is no standard legal framework for endolysin applications since this method has specific requirements and guidelines.

Endolysins are still largely under research and development, particularly in the context of clinical trials and regulatory approvals. Their progression through clinical trials and regulatory approval processes can be complex and time-consuming. To date, Nomad Bioscience GmbH submitted a GRAS notice (GRN 000802) to the FDA for an endolysin preparation. The notice included safety evaluation data from their own studies showing the endolysin’s bactericidal effects on lab and cooked meat tests against *C. perfringens*. Nomad set criteria for activity, stability, and nicotine/anabasine levels. Estimated dietary exposure was 2.6 mg/person/day based on a 10 mg/kg application rate (https://www.nomadbioscience.com/intellectual-property-publications/, accessed on 5 June 2023). FDA reviewed the data and determined the endolysin preparation safe for its intended use.

## 8. Fourth Industrial Revolution: Digital Innovations in Food Industry

Digital technologies such as AI, big data analytics, internet of things (IoT), and blockchain, as well as other technological advances such as smart sensors, robotics, digital twins, and cyber–physical systems are playing role in food industry. These technologies are part of the fourth industrial revolution (industry 4.0) and have significantly modified the food industry, leading to substantial consequences for the environment, economics, and human health [111,112].

Industry 4.0 technologies can improve the food industry in several ways. These technologies could promote extensive digital transformation of everything possible and sustainable development along the different stages of the food value chain, saving time and reducing cost [113]. For example, the use of hyperspectral sensors based on different spectroscopic principles to optimize and monitor at any time and stage multiple processing conditions throughout the course of an enzymatic hydrolysis process for various food by-products [114]. These “green” technologies would reduce food waste, and give opportunities to customize food products and obtain desirable products with specific quality attributes. Consequently, it becomes possible to increase profitability, reduce food wastes, optimize customer needs, and increase consumer satisfaction [115]. The industry 4.0 technologies will contribute to the green transition toward more sustainable, intelligent, innovative food production systems, with improved efficiency and productivity.

Industry 4.0 technologies offer valuable tools for advancing hybrid endolysin development through synthetic engineering. Currently, most of the biomolecular, functional, and structural studies of holins and spanins are examined using *E. coli* expression systems, but this is challenging due to toxicity. Recent advances in protein synthesis and AI and ML approaches are helping to overcome this challenge [116]. AI and ML can predict effective endolysin components such as holins [116], while IoT and robotics optimize experimental conditions. Blockchain ensures data integrity, and digital twins model interactions [117]. Advanced manufacturing creates precise structures, and high-throughput methods rapidly screen variants. Collaborative platforms facilitate global teamwork [116]. A machine learning approach was developed to predict PVPsusing phage protein sequences. Established machine learning methods, including basic and ensemble techniques, were utilized with protein sequence composition features for PVP prediction. The gradient boosting classifier method demonstrated the highest accuracy, achieving 80% on the training dataset and 83% on an independent dataset—outperforming other methods. A user-friendly web server, created for PVP prediction from phage protein sequences, is accessible for free. This tool has the potential to aid large-scale PVP prediction and guide experimental study design based on hypotheses [118]. By harnessing these technologies, hybrid endolysins can be efficiently designed, tested, and optimized, paving the way for novel antimicrobial solutions.

However, there are several challenges associated with implementing these technologies in the food industry. The adoption of industry 4.0 elements by the food industry is not without challenges. For example, security and privacy issues when collecting large amounts of data over time, makes them more vulnerable to confidentiality attacks. Setting common standards and legal frameworks, as well as establishing the proper regulatory environment, is important to ensure the protection and consistency of data, especially with cross-border data flows. Most of the emerging technologies are still confined to laboratory-scale experiments and are not commercially available because of the gap between laboratory-scale research and real-time applications. Moreover, lack of technical and technological skills is another issue that hinders wider acceptance of industry 4.0 and its new technologies and innovations. Other barriers may be related to specific technologies. Although successful applications of AI, ML, and big data analytics have been reported both for specific operations and along the food value chain, adoption of these technologies is still limited. Barriers are related to challenges with data (infrastructure, quality, standardization, security, and privacy), cost, and lack of adaptability to an industrial environment.

## 9. Conclusions

The utilization of phage endolysins, enzymatic compounds originating from bacteriophages, presents a promising avenue for enhancing food safety within the realm of food processing systems. These enzymes exhibit the capability to effectively degrade bacterial cell walls, thereby offering a means to regulate pathogens and bolster food preservation. This comprehensive review has underscored the potential of endolysins in safeguarding food products, examining both native and engineered variants for bacterial control throughout the food production chain.

Nonetheless, the analysis reveals a challenge in the form of time-consuming and resource-intensive in vitro methods for characterizing endolysins. In response, the emergence of advanced high-throughput sequencing technology emerges as a pivotal opportunity. It calls for the establishment of a robust computational framework, effectively streamlining the characterization of newly identified endolysins. This forward-looking approach, aligned with the industry 4.0 paradigm, hinges on the potency of machine learning techniques. These tools possess the capacity to dissect intricate datasets and facilitate pattern recognition, thereby accelerating the advancement of endolysin research. By embracing these strategies, there is a clear potential to amplify food safety measures, tailor solutions, enhance efficiency, and foster collaboration. Simultaneously, regulatory challenges can be mitigated, leading to the timely availability of improved products that align with evolving consumer demands and industry standards.

## Figures and Tables

**Figure 1 cells-12-02169-f001:**
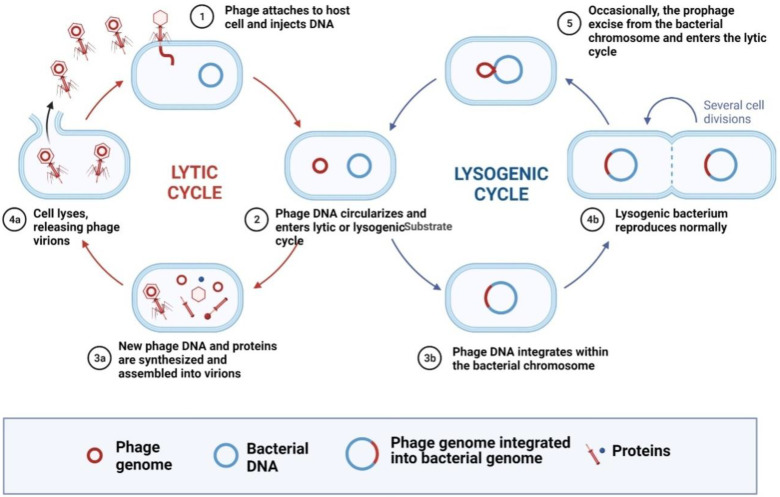
Graphic representation lytic and lysogenic cycle of a phage along with the various protein produced during the phage host interaction.

**Figure 2 cells-12-02169-f002:**
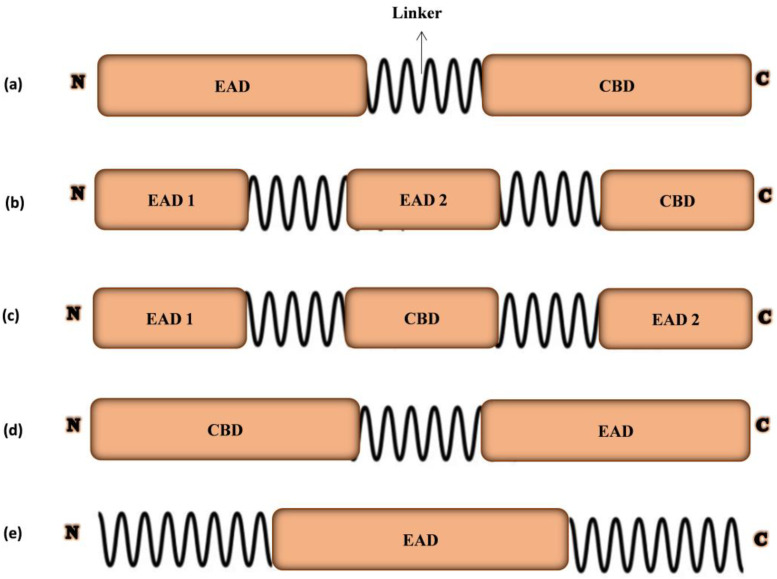
Modular configuration models of common phage endolysins. (**a**) Model with one N-terminal enzymatically active domain (EAD) and a C-terminal cell wall-binding domain (CBD). (**b**) Multi-domain model with two EADs and a C-terminal CBD. (**c**) Multi-domain model with a CBD located between two EADs. (**d**) Modular endolysin with a C-terminal EAD and an N-terminal CBD. (**e**) Simple globular model of an EAD with no CBD.

**Figure 3 cells-12-02169-f003:**
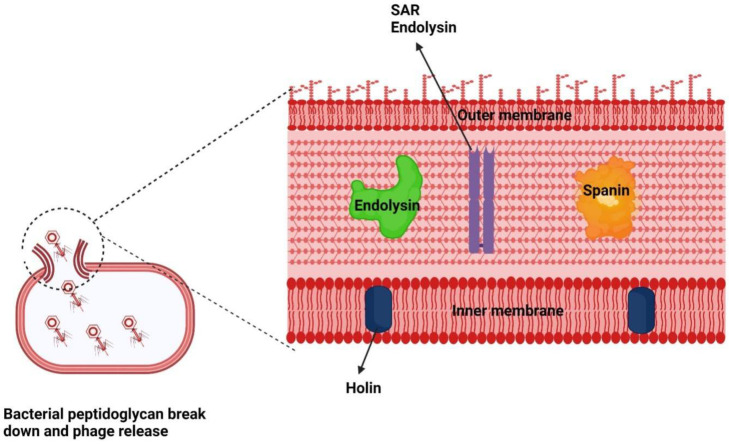
Mode of action of an endolysin.

**Figure 4 cells-12-02169-f004:**
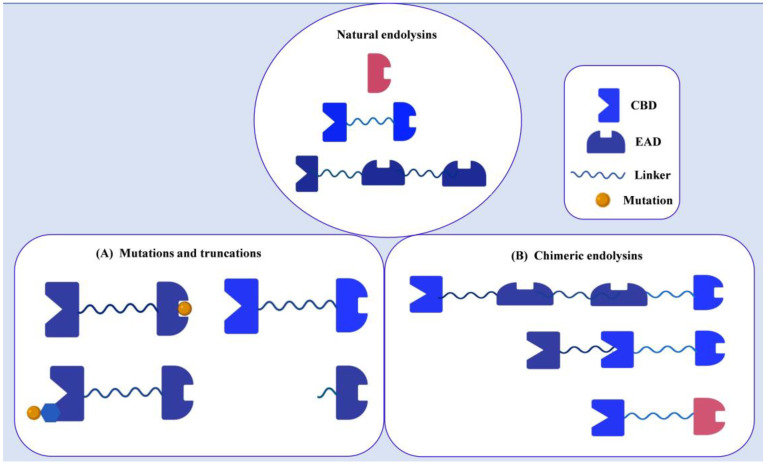
Engineering strategies used to modify endolysins. Natural endolysins are composed of a single EAD, or they exhibit a modular arrangement including at least one EAD and one CBD. Building upon this fundamental structure, numerous modifications have been generated through protein engineering. (**A**) Enhancements in activity and modifications in specificity have been achieved through mutagenesis and the construction of truncations. (**B**) By engaging in domain swapping, chimeric endolysins have been formed, combining CBDs and EADs from distinct endolysins.

**Figure 5 cells-12-02169-f005:**
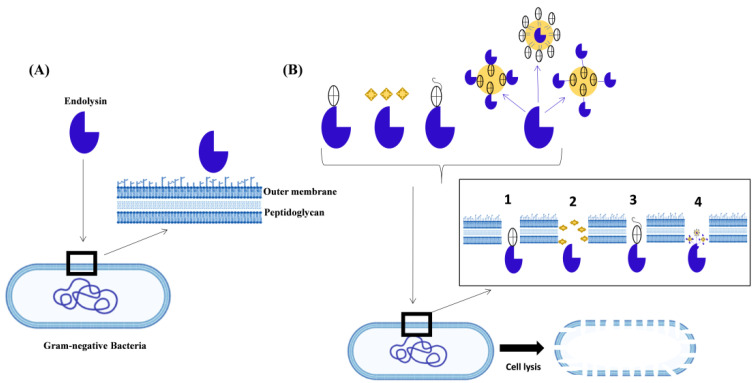
Strategies for endolysins to cause cell lysis of Gram-negative bacteria: (**A**) Natural endolysins cannot induce lysis in Gram-negative bacteria through an external mechanism due to the presence of the outer membrane (OM), which acts as a barrier preventing direct contact with the peptidoglycan. (**B**) (1) Identifying lysins with intrinsic OM permeability; (2) using in combination with OMPs; (3) engineering lysins with OM disrupting properties; (4) formulating lysins into OM-penetrating carrier systems.

**Figure 6 cells-12-02169-f006:**
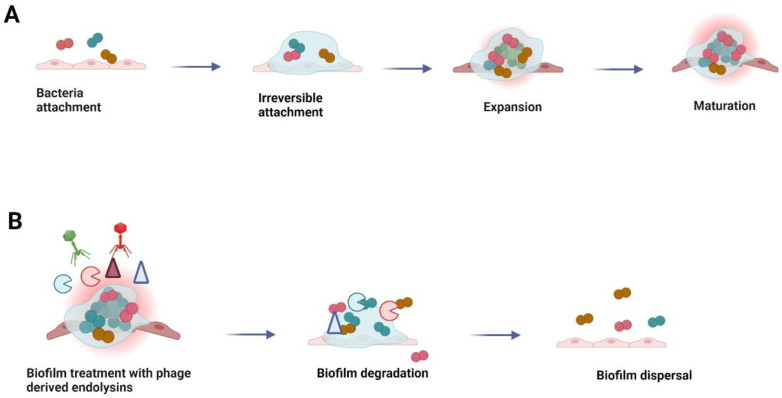
Biofilm growth stages and degradation: (**A**) The stages of bacterial biofilm formation from the initial attachment to mature biofilm formation; (**B**) process of biofilm dispersal using phage endolysins as agents of biofilm degradation and bacterial cell lysis.

**Figure 7 cells-12-02169-f007:**
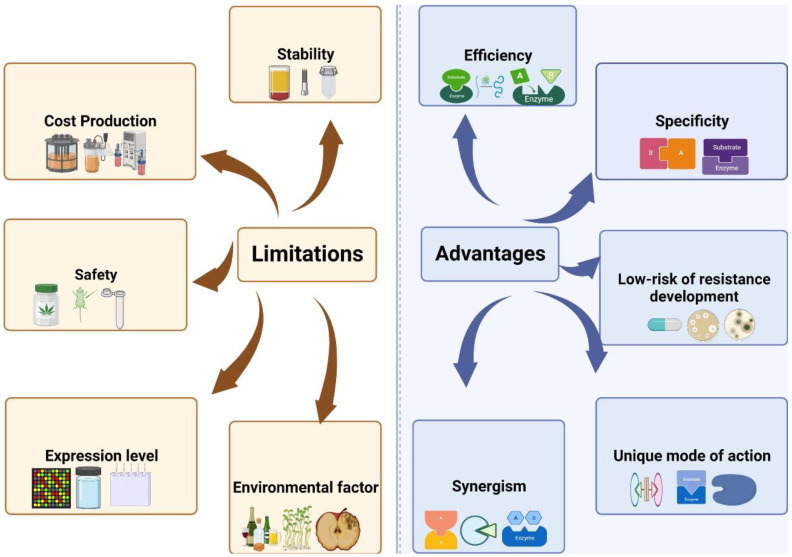
Advantages and limitations of endolysins in food industry. Phage endolysins are specific for their target with unique mode of action and have low risk of resistance development. However, safety and stability of endolysins is a big challenge as well as their cost production.

## Data Availability

Not applicable.

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
