# Peer review of "Phage Endolysins: Advances in the World of Food Safety"

_cells, 2023, doi:10.3390/cells12172169_

Round 1

Reviewer 1 Report

-          The review discusses the use of endolysins of phage in biocontrol against food-borne pathogens to enhance food security. The manuscript is well-organised and provides information of interest to many readers and researchers. Only a few points could be suggested as follows:

-          Keywords list should be revised as this list should contain different words than those present in the title

-          The figures are well-presented, but the font size should be the same in all figures. For example, it seems that Figure 2 has bold and bigger text size than the other figures

-          By the end of the introduction, the novelty of this manuscript compared to other recently published reviews should be highlighted more. The authors only mentioned publications from the years 2010 and 2011, while more recent references should be added.

-          There are too many abbreviations in this manuscript. I think there is no need to use abbreviations (such as GEWL) if these are not employed in the following sections. Please check the whole manuscript in order to fix this issue.

-          Can recent advanced technologies of Industry 4.0 (such as artificial intelligence, smart sensors…. Please check the following reference for more details: https://doi.org/10.1080/10408398.2022.2034735) have a role to play in the development of hybrid endolysins by synthetic engineering?

The English is fine

Author Response

Reviewer1

The review discusses the use of endolysins of phage in bio-control against food-borne pathogens to enhance food security. The manuscript is well-organized and provides information of interest to many readers and researchers. Only a few points could be suggested as follows:

-          Keywords list should be revised as this list should contain different words than those present in the title

Answer: Thank you for your valuable suggestion. We have revised the keyword list as per your suggestion. 

-          The figures are well-presented, but the font size should be the same in all figures. For example, it seems that Figure 2 has bold and bigger text size than the other figures.

Answer: Revised as per your suggestion

-          By the end of the introduction, the novelty of this manuscript compared to other recently published reviews should be highlighted more. The authors only mentioned publications from the years 2010 and 2011, while more recent references should be added.

Answer: Revised as per your suggestion

-          There are too many abbreviations in this manuscript. I think there is no need to use abbreviations (such as GEWL) if these are not employed in the following sections. Please check the whole manuscript in order to fix this issue.

Answer: Revised as per your suggestion

-          Can recent advanced technologies of Industry 4.0 (such as artificial intelligence, smart sensors…. Please check the following reference for more details: https://doi.org/10.1080/10408398.2022.2034735) have a role to play in the development of hybrid endolysins by synthetic engineering?

Answer: We are thankful for your valuable suggestion. Manuscript has been updated according to your suggestion.

Reviewer 2 Report

Review comments

The manuscript titled “Phage Endolysins: Advances in the World of Food Safety” by Nazir et al., reviews the role of phage endolysins in combating bacterial infections with especial focus on food infections. The review is informative.

However, I have certain important concerns as given below. My comments are highlighted in bold letters after authors’ review text.

Abstract:

·         “The endolysins of phages (peptidoglycan hydrolases) are helpful in several fields for controlling bacterial contamination”

·         The sentence is ambiguous! What does “fields” stand for?

·         A review on similar topic has been published in the year 2016 with the title “Bacteriophage endolysins: applications for food safety”

·         The review also started with similar frame of sentence that is “Bacteriophage endolysins (peptidoglycan hydrolases) have emerged as a new class of antimicrobial agents useful for controlling bacterial infection or other unwanted contaminations in various fields...”

·         Authors can think of a different start.

1. Introduction

·         “It is most appropriate to term these proteins as 'lysins' since they have functions both inside and outside the cell, but the two terms are interchangeable in the field. At the same time naturally, endolysins lyse the host cell internally (Figure 1)”.

·         However, intended for Gram-positive bacteria, endolysins can act as exolysins by degrading their outer peptidoglycan layer. Still, they cannot degrade the outer membrane in the case of Gram-negative bacterial cells

Ø  Above sentence frames are confusing. Please clarify.

·         “Endolysins from Gram-positive phage hosts are typically categorized…”

Ø  How can endolysins be derived from gram positive phage host? Clarification in the text would add value for the readers.

Ø  As references, author has citated many review articles on similar topic, whereas citation of the original research article should be given [eg. EADs provide the simple enzymatic action that degrades the peptidoglycan structure, while CBD can identify and bind to specific cell wall linked ligand molecules with great specificity [14]. Here, Reference 14 is another review article on application of phage endolysin in food safety. 

·         Although good hygiene practices have been practiced but alarmingly, food-borne diseases have elevated and outbreaks are not decreasing.

Ø  Along with good hygiene practice, good manufacturing practice should also be highlighted.

·         “However, even though phage applications have some serious concerns such as gene transduction and spreading phage resistant bacteria, amazingly, endolysins do not cause such problems [19]”

Ø  Article cited as Reference [19] does not contain information on phage endolysin.

2. Primary features of endolysins

2.1. Structure and enzymatic activity of endolysins

·         “Gram-negative phages can have various structures in their endolysin...”

Ø  Can phages infectious to gram negative bacteria be called as gram negative phage?

2.2. Endolysins: mode of action

·         “Previously, it was an assumption that the holin–endolysin system of Gram-positive phages allows endolysin to damage the bacterial cell wall [32]. However, new studies have found another protein named spanin also required for the breakdown of outer membrane [33]”

Ø  From the sentence it appears that spanin is only found in the phages infecting the gram-positive bacteria, however spanin can also be coded by bacteriophages infecting gram negative bacteria.

3. Impact of Endolysins as Bio-control agents in the food industry

·         “Numerous research describes the characterization of new endolysins against foodborne pathogens when employed to food and utensils. Ref. [41]”

Ø  This reference cites the preclinical Safety Studies of Pneumococcal Endolysins Cpl-1 and Pal. It is a single preclinical trial study.

·         “Furthermore, a few studies have shown these endolysins' safety parameters in the human health. Ref [42]”

Ø  Again this reference cites a single preclinical safety study; “Preclinical safety evaluation of intravenously administered SAL200 containing the recombinant phage endolysin SAL-1 as a pharmaceutical ingredient it is a single safety study”

4. Development of hybrid endolysins by synthetic engineering to combat food-borne pathogens

·         “Phages could not unilaterally annihilate the bacteria during co-evolution as they need them for reproduction. Therefore, it is a fact that endolysins have not evolved to be maximally active.”

Ø  The statement is interesting, but do phages intrinsically sense the need of bacterial population for their reproduction, thus existence? Or it is bacteria that evolve itself to survive from the phage scavenging by modifying its structural components. As lytic phage could lyse single bacterium very efficiently within minutes, however it is bacterial evolutionary paradigm that makes the bacteria thrive against phages by acquiring phage resistance. The presence of sense of coexistence in phages requires extensive studies. Endolysins available with phages are potent enough to lyse a bacterium, it is the bacterial survival pressure that makes them recalcitrant to phages.

·         Figure 5

Ø   Figure 5 should be more transparent;

- all the four images should be properly numbered.

- explained the figure should be clearly explained

- figure with subfigure numbering should be mentioned in the text for better  

    understanding

- Endolysin is composed of CBD  and EAD  and linked with the linker,

What is the natural orientation of the linkage and how it is impacting the activity of the endolysin? If this  is the orientation than what is this structure stands for? If it is a natural endolysin then according to above mentioned structures and orientation the natural endolysin should be of this structure.

What is the significance and activity of this orientation?

Ø  Discussion on “Endolysins with intrinsic membrane-passing capabilities” should be separated from the subheading “4.2. Engineering of endolysins against Gram-negative bacteria” as the subheading implies structural manipulation of the endolysin

·         “Horgan et al. constructed synthetic derivatives of Staphylococcus LysK endolysin and discovered CHAPk had a significantly higher lytic activity when used in vitro against Staphylococci than the natural form of LysK”

Ø  What is CHAPk?

8. Conclusion

·         Authors have mentioned in the conclusion “Endolysins are potentially influential enzymes that could prevent food-borne infections and enhance safety in the food and pharmaceutical industry”

Ø  No study on the use of the endolysin in pharmaceutical products has been mentioned in the review.

General Comments.

Ø  Effect of endolysin on food and its nutritional values if any should be included.

Ø  Status on Ongoing Clinical trails and approval status of the endolysin should be included.

Ø  Sentence framings can be improved.

Minor corrections maybe required

Author Response

Reviewer 2

I reviewed the manuscript titled “Phage Endolysins: Advances in the World of Food Safety. The manuscript is well written with an innovative idea. However, some sections of this review must be improved by including latest scientific literature

Abstract

Review findings should be introduced after the review objectives

Answer: Revised as per your suggestion

Conclusions and recommendations should be included in the abstract

Answer: Revised and incorporated required data in manuscript

Figure 1. quality must be improved. The figure was created with Biorender (http://bioren- 33 der.com)…. Can be removed

Answer: Revised as per your suggestion

Need of conducting this review must be highlighted in introduction

Answer: Revised as per your suggestion

More recent relevant literature must be cited in the introduction section

Answer: Revised as per your suggestion

Is figure authors their own construction or taken from literature? If it is from literature, please provide a copyright permission and statement

Answer: All figures constructed by own.

Since food safety is central to this review, authors must provide in-depth discussion on section 3. As such, this section is superficial. For example, authors limited the discussion to milk and/or very few meat products. I suggest improving in this area. For example, authors can divide into sections Fruits and vegetables, Milk and milk products, Meat and poultry products, During storage (in general) of all above

How the endolysins showing the impact on different bacteria in the food industry (Fruits and vegetables; Milk and milk products; Meat and poultry products)

Answer: This section has been revised as per your suggestion.

Section 4 is well written

All scientific names must be in Italics

Answer: Revised as per your suggestion

Section 7 : UNIQUE can be removed

Answer: Revised as per your suggestion

Conclusions must be revised to reflect the review findings. Recommendations and limitations must be highlighted in section 8

Answer: Revised as per your suggestion

References are not according to the journal format. Please revise it 

Answer: Revised

Reviewer 3 Report

I reviewed the manuscript titled “Phage Endolysins: Advances in the World of Food Safety. The manuscript is well written with an innovative idea. However, some sections of this review must be improved by including latest scientific literature

Abstract

Review findings should be introduced after the review objectives

Conclusions and recommendations should be included in the abstract

Figure 1. quality must be improved. The figure was created with Biorender (http://bioren- 33 der.com)…. Can be removed

Need of conducting this review must be highlighted in introduction

More recent relevant literature must be cited in the introduction section

Is figure authors their own construction or taken from literature? If it is from literature, please provide a copyright permission and statement

Since food safety is central to this review, authors must provide in-depth discussion on section 3. As such, this section is superficial. For example, authors limited the discussion to milk and/or very few meat products. I suggest improving in this area. For example, authors can divide into sections

Fruits and vegetables

Milk and milk products

Meat and poultry products

During storage (in general) of all above

How the endolysins showing the impact on different bacteria in the food industry (Fruits and vegetables; Milk and milk products; Meat and poultry products)

Section 4 is well written

All scientific names must be in Italics

Section 7 : UNIQUE can be removed

Conclusions must be revised to reflect the review findings. Recommendations and limitations must be highlighted in section 8

References are not according to the journal format. Please revise it 

Author Response

Reviewer 3

abstract section

-        The objective was not clear please re write it

Answer: Revised

-        Authors should stress the novelty of this work

Answer: Revised

-        2 -3 concise and conclusive sentences should be added at the end of the abstract section

Answer: Revised as per your suggestion

  • Introduction section

-        please check English language, some sentences should be well revised

Answer: Revised as per your suggestion

-        How about the practical applications of phages in food industry? This point should be discussed deeply

Answer: Required data has been incorporated in the manuscript

-        What kind of industry field, phages are used- Some examples should be developed

Answer: The required literature have been added in the manuscript

-        Some mechanism of action against bacteria should be discussed

Answer: We are thankful for your valuable suggestion. Manuscript revised as per your suggestion.

-        - L54-55 this sentence deserved some further details

Answer: Required data incorporated in the manuscript

  • The strategy followed by the authors to write this review should be developed in subsection after Introduction part

Answer: Revised as per your suggestion

  • All abbreviation should be grouped in a sub section “Abbreviations”

Answer: Revised as per your suggestion

  • The section 2 should be concise and avoid to use general data

Answer: We have summarized this section and updated the manuscript as per your suggestion.

  • Regarding Table 1, the column of “Results” should be discussed

Answer: Manuscript has been updated according to your suggestion

  • All tables should be discussed

Answer: Revised as per your suggestion

  • Please improve the conclusion section

Answer: Revised

Reviewer 4 Report

·       Abstract section

-        The objective was not clear please re write it

-        Authors should stress the novelty of this work

-        2 -3 concise and conclusive sentences should be added at the end of the abstract section

·       Introduction section

-        please check English language, some sentences should be well revised

-        How about the practical applications of phages in food industry? This point should be discussed deeply

-        What kind of industry field, phages are used

-        Some examples should be developed

-        Some mechanism of action against bacteria should be discussed

-        - L54-55 this sentence deserved some further details

·       The strategy followed by the authors to write this review should be developed in subsection after Introduction part

·       All abbreviation should be grouped in a sub section “Abbreviations”

·       The section 2 should be concise and avoid to use general data

·       Regarding Table 1, the column of “Results” should be discussed

·       All tables should be discussed

·       Please improve the conclusion section

Author Response

Review comments

The manuscript titled “Phage Endolysins: Advances in the World of Food Safety” by Nazir et al., reviews the role of phage endolysins in combating bacterial infections with especial focus on food infections. The review is informative.

However, I have certain important concerns as given below. My comments are highlighted in bold letters after authors’ review text.

Abstract:

ï‚· “The endolysins of phages (peptidoglycan hydrolases) are helpful in several fields for controlling bacterial contamination”

ï‚· The sentence is ambiguous! What does “fields” stand for?

Answer: Sentence has been revised.

ï‚· A review on similar topic has been published in the year 2016 with the title “Bacteriophage endolysins: applications for food safety”

ï‚· The review also started with similar frame of sentence that is “Bacteriophage endolysins (peptidoglycan hydrolases) have emerged as a new class of antimicrobial agents useful for controlling bacterial infection or other unwanted contaminations in various fields...”

ï‚· Authors can think of a different start.

Answer: Dear editor, We have revised the sentence as per your suggestion.

  1. Introduction

ï‚· “It is most appropriate to term these proteins as 'lysins' since they have functions both inside and outside the cell, but the two terms are interchangeable in the field. At the same time naturally, endolysins lyse the host cell internally (Figure 1)”.

ï‚· However, intended for Gram-positive bacteria, endolysins can act as exolysins by degrading their outer peptidoglycan layer. Still, they cannot degrade the outer membrane in the case of Gram-negative bacterial cells

 Above sentence frames are confusing. Please clarify.

Answer: We have revised these statements.

ï‚· “Endolysins from Gram-positive phage hosts are typically categorized…”

 How can endolysins be derived from gram positive phage host? Clarification in the text would add value for the readers.

Answer: Required data incorporated in the manuscript

 As references, author has citated many review articles on similar topic, whereas citation of the original research article should be given [eg. EADs provide the simple enzymatic action that degrades the peptidoglycan structure, while CBD can identify and bind to specific cell wall linked ligand molecules with great specificity [14]. Here, Reference 14 is another review article on application of phage endolysin in food safety.

Answer: Reference has updated.

Although good hygiene practices have been practiced but alarmingly, food-borne diseases have elevated and outbreaks are not decreasing.

 Along with good hygiene practice, good manufacturing practice should also be highlighted.

Answer: Revised as per your suggestion

ï‚· “However, even though phage applications have some serious concerns such as gene transduction and spreading phage resistant bacteria, amazingly, endolysins do not cause such problems [19]”

Article cited as Reference [19] does not contain information on phage endolysin.

Answer: We have updated this reference.

  1. Primary features of endolysins

2.1. Structure and enzymatic activity of endolysins

ï‚· “Gram-negative phages can have various structures in their endolysin...”

 Can phages infectious to gram negative bacteria be called as gram negative phage?

Answer: We are sorry for confusion. We have made corrections.

2.2. Endolysins: mode of action

ï‚· “Previously, it was an assumption that the holin–endolysin system of Gram-positive phages allows endolysin to damage the bacterial cell wall [32]. However, new studies have found another protein named spanin also required for the breakdown of outer membrane [33]”

 From the sentence it appears that spanin is only found in the phages infecting the gram-positive bacteria, however spanin can also be coded by bacteriophages infecting gram negative bacteria.

Answer: Thank you for pointing out. Sentence has been revised.

  1. Impact of Endolysins as Bio-control agents in the food industry

ï‚· “Numerous research describes the characterization of new endolysins against foodborne pathogens when employed to food and utensils. Ref. [41]”

 This reference cites the preclinical Safety Studies of Pneumococcal Endolysins Cpl-1 and Pal. It is a single preclinical trial study.

Answer: Thank you for pointing out. This has been revised.

ï‚· “Furthermore, a few studies have shown these endolysins' safety parameters in the human health. Ref [42]”

 Again this reference cites a single preclinical safety study; “Preclinical safety evaluation of intravenously administered SAL200 containing the recombinant phage endolysin SAL-1 as a pharmaceutical ingredient it is a single safety study”

Answer: Manuscript has updated.

  1. Development of hybrid endolysins by synthetic engineering to combat food-borne pathogens

ï‚· “Phages could not unilaterally annihilate the bacteria during co-evolution as they need them for reproduction. Therefore, it is a fact that endolysins have not evolved to be maximally active.”

The statement is interesting, but do phages intrinsically sense the need of bacterial population for their reproduction, thus existence? Or it is bacteria that evolve itself to survive from the phage scavenging by modifying its structural components. As lytic phage could lyse single bacterium very efficiently within minutes, however it is bacterial evolutionary paradigm that makes the bacteria thrive against phages by acquiring phage resistance. The presence of sense of coexistence in phages requires extensive studies. Endolysins available with phages are potent enough to lyse a bacterium, it is the bacterial survival pressure that makes them recalcitrant to phages.

Revised

Phages do not intrinsically "sense" the need of bacterial populations for their reproduction or existence in a conscious manner. Instead, their interactions with bacteria are the outcome of evolutionary processes driven by genetic and biochemical interactions. Bacteria can evolve mechanisms to resist phages, and the coexistence of these entities involves intricate dynamics that continue to be a subject of extensive research. In essence, even though endolysins are potent and efficient, bacteria can evolve countermeasures to resist their action in order to ensure their own survival. This ongoing evolutionary arms race between phages and bacteria has led to the development of complex interactions and strategies on both sides. This statement succinctly underscores the dynamic and nuanced nature of the phage-bacteria relationship.

ï‚· Figure 5

 Figure 5 should be more transparent; all the four images should be properly numbered.

Revised as per your suggestion

- explained the figure should be clearly explained

Figure legends has revised  

- figure with subfigure numbering should be mentioned in the text for better

understanding

Answer: We have revised it according to your instructions.

- Endolysin is composed of CBD and EAD and linked with the linker,

What is the natural orientation of the linkage and how it is impacting the activity of the endolsin?

Revised

The inherent alignment of functional domains within endolysin proteins is often referred to as the natural orientation of linkages. The implications of this natural arrangement of domains within endolysins can be quite substantial. The specific configuration and sequential arrangement of these domains play a pivotal role in determining how effectively the endolysin is able to identify and break down the bacterial cell wall. One of the most noteworthy characteristics of numerous endolysins is their modular structure, which offers a disruptive quality, opening avenues to tailor endolysins for specific applications. This modular nature allows for the swapping of domains from different endolysins, offering a means to adjust specificity, amplify activity, or enhance solubility to align with particular needs or objectives.

 If this is the orientation than what is this structure stands for?

If it is a natural endolysin then according to above mentioned structures and orientation the natural endolysin should be of this structure

Answer: We are sorry for the confusion. We have modify figure and figure legends in well mannered.

What is the significance and activity of this orientation?

Usually, modular endolysins have one or two EADs and one CBD linked by a flexible region called the linker. Detailed explanation under the section 2.

 Discussion on “Endolysins with intrinsic membrane-passing capabilities” should be separated from the subheading “4.2. Engineering of endolysins against Gram-negative bacteria” as the subheading implies structural manipulation of the endolysin

Answer: We can modify endolysins in four different ways to break the cell wall of gram negative bacteria. However, we have mentioned these techniques with numbers accordingly.

ï‚· “Horgan et al. constructed synthetic derivatives of Staphylococcus LysK endolysin and discovered CHAPk had a significantly higher lytic activity when used in vitro against Staphylococci than the natural form of LysK”

 What is CHAPk?

Required data incorporated in the manuscript.

Horgan et al. constructed modified version of the LysK (an endolysin from staphylococcal phage K), featuring an N-terminal CHAPK domain, a central amidase domain, and a C-terminal SH3b cell wall-binding domain. This design enables CHAPK to break down bacterial peptidoglycan between the tetra-peptide stem and the penta-glycine bridge. Notably, CHAPK exhibited notably greater lytic activity against Staphylococci in vitro compared to the natural form of LysK.

  1. Conclusion

ï‚· Authors have mentioned in the conclusion “Endolysins are potentially influential enzymes that could prevent food-borne infections and enhance safety in the food and pharmaceutical industry”

 No study on the use of the endolysin in pharmaceutical products has been mentioned in the review.

Answer: Thank you for mentioning. We are trying to focus on use of endolysins regarding food safety. However, sentence has been revised.

General Comments.

 Effect of endolysin on food and its nutritional values if any should be included.

Answer: We have tried our best but we couldn’t find any article about the effect of endolysin on food and its nutritional values yet.

 Status on Ongoing Clinical trials and approval status of the endolysin should be included.

Revised as per your suggestion.

Answer: Endolysins are still largely under research and development, particularly in the context of clinical trials and regulatory approvals. Their progression through clinical trials and regulatory approval processes can be complex and time-consuming. To date, Nomad Bioscience GmbH submitted a Generally Recognized As Safe (GRAS) notice (GRN 000802) to the FDA for an endolysin preparation. The notice included safety evaluation data from their own studies showing the endolysin's bactericidal effects on lab and cooked meat tests against C. perfringens. Nomad set criteria for activity, stability, and nicotine/anabasine levels. Estimated dietary exposure was 2.6 mg/person/day based on a 10 mg/kg application rate. FDA reviewed the data and determined the endolysin preparation safe for its intended use.

 Sentence framings can be improved.

Answer: Revised as per you suggestion throughout the manuscript